



# Development of a regional feature selection-based machine learning system (RFSML v1.0) for air pollution forecasting over China

Li Fang[1], Jianbing Jin*[1], Arjo Segers[2], Hai Xiang Lin[3,4], Mijie Pang[1], Cong Xiao[5], Tuo Deng[4], and Hong Liao*[1]

[1]Jiangsu Key Laboratory of Atmospheric Environment Monitoring and Pollution Control, Jiangsu Collaborative Innovation Center of Atmospheric Environment and Equipment Technology, School of Environmental Science and Engineering, Nanjing University of Information Science and Technology, Nanjing, Jiangsu, China
[2]TNO, Department of Climate, Air and Sustainability, The Netherlands
[3]Institute of Environmental Sciences, Leiden University, The Netherlands
[4]Delft Institute of Applied Mathematics, Delft University of Technology, Delft, the Netherlands
[5]Key Laboratory of Petroleum Engineering, Ministry of Education, China University of Petroleum, Beijing, China

**Correspondence:** Jianbing Jin (jianbing.jin@nuist.edu.cn) and Hong Liao (hongliao@nuist.edu.cn)

**Abstract.** With the explosive growth of atmospheric data, machine learning models have achieved great success in air pollution forecasting because of their higher computational efficiency than the traditional chemical transport models. However, in previous studies, new prediction algorithms have been tested only at stations or in a small region; a large-scale air quality forecasting model remains lacking to date. Huge dimensionality also means that redundant input data may lead to increased complexity

and therefore the over-fitting of machine learning models. Feature selection is a key topic in machine learning development, but it has not yet been explored in atmosphere-related applications. In this work, a regional feature selection-based machine learning (RFSML) system was developed, which is capable of predicting air quality in the short-term with high accuracy at the national scale. Ensemble-Shapley additive global importance analysis is combined with the RFSML system to extract significant regional features and eliminate redundant variables at an affordable computational expense. The significance of the

regional features is also explained physically. Compared with a standard machine learning system fed with relative features, the RFSML system driven by the selected key features results in superior interpretability, less training time, and more accurate predictions. This study also provides insights into the difference in interpretability among machine learning models (i.e., random forest, gradient boosting, and multi-layer perceptron models).

## 1 Introduction

With ongoing economic development and modern industrialization, the subsequent air pollution poses serious threats to resident health (Liu and Diamond, 2005; Li et al., 2014). After tobacco and high blood pressure, air pollution has ranked third in risk factors for death and disability in China over the past few decades (Murray et al., 2020). The primary air pollutants in China are particulate matter (PM), sulfur dioxide ($SO_2$), carbon monoxide (CO), nitrogen oxides ($NO_x$) and ozone ($O_3$) (Song et al., 2017b). $PM_{2.5}$ or respirable PM in air with an aerodynamic diameter below $2.5\mu m$ is the primary air pollutant, and it

has attracted considerable attention from researchers (Zhai et al., 2019). Exposure to either long-term or short-term $PM_{2.5}$ is



related to respiratory symptoms, lung disease, cardiovascular disease, premature death, and other adverse health effects (Pui et al., 2014; Di et al., 2019). Burnett et al. (2018) and Song et al. (2017a) reported that $PM_{2.5}$ pollution in winter, particularly in northern China, is severe. It accounted for 15.5% (1.7 million) of all deaths in China in 2015, despite an improvement in air quality since 2013. In recent studies, the global exposure mortality model has estimated that 140,200 premature deaths

from 2015 to 2019 can be attributed to long-term exposure to $PM_{2.5}$ (Hao et al., 2021). An accurate air quality forecast (e.g., forecasting $PM_{2.5}$) is therefore valuable to policy makers and health professionals for epidemiological control (Xue et al., 2019). In addition, it can provide an early warning for residents, particularly for children, the elderly, and people with respiratory or cardiovascular problems (Hu et al., 2017).

The development of an air pollution forecasting model is possible, as atmospheric chemistry and physical rules have

been explored and are now understood in depth (Sun and Li, 2020b). In addition, our ever-increasing computational power can support the complex and heavy computational tasks required for this type of model (Reichstein et al., 2019). Deterministic models, such as chemical transport models (CTMs), and data-driven methods are commonly employed in forecasting (Cobourn, 2010; Xu et al., 2021). In several studies, air pollution forecasting has been performed using mainstream air quality CTMs, such as the Weather Research and Forecasting model with chemistry (Grell et al., 2005; Zhou et al., 2017), Community Multi-

scale Air Quality with chemistry model (Liu et al., 2018), and GEOS-Chem (Bey et al., 2001; Jeong and Park, 2018). These CTMs can reproduce real atmospheric situations (Hutzell and Luecken, 2008; Shtein et al., 2020); however, they exhibit several shortcomings. One of the most difficult setbacks is the high uncertainty in emission inventories (Huang et al., 2021), which is a great challenge given the variety of contributing sources, complexity of the spatial-temporal profiles, and lack of reliable in situ measurements (Li et al., 2017a). Additionally, an idealistic deterministic model requires a delicate and thorough understanding

of physical and chemical processes in the atmosphere (Sun and Li, 2020a) and an enormous computational capacity to resolve fine-scale variabilities. Therefore, CTMs alone have failed to meet the requirements for an effective air quality early warning system.

In contrast to CTMs, data-driven methods that do not require a profound knowledge of the complex composition or structure of the atmosphere are also widely utilized in atmospheric modeling (Ziomas et al., 1995; Xi et al., 2015). Many of

these methods have been employed for air pollution forecasting, including multiple linear regression (Sawaragi et al., 1979), nonlinear regression models such as principal component regression (Shishegaran et al., 2020), hidden Markov models (Sun et al., 2013), support vector machine (Abu Awad et al., 2017), and artificial neural networks (Fernando et al., 2012). Of these methods, machine learning models have gained the greatest popularity because of their capacity to learn complex and nonlinear relationships by assimilating "big" training datasets (Masih, 2019; Leufen et al., 2021). Machine learning has brought great

opportunities and challenges to the geophysical research community (Yu and Ma, 2021).

With the explosive growth of data in earth science, the superiority of machine learning for massive data applications has become increasingly prominent (Reichstein et al., 2019). The most representative example is to perform predictions for a target site using a machine learning model trained via a long-term series of in situ historical measurements. The China Ministry of Environmental Protection (MEP) has established many ground-based stations measuring the primary pollutants since 2013

(Zhai et al., 2019). At present, the monitoring network comprises more than 1500 field stations covering all of China, as can





be seen in Fig. 1. The richness of air quality observations from the monitoring network provides valuable training data and stimulates the development of machine learning air quality forecasting in China (Xu et al., 2021). Previous studies on air pollution forecasting in China have utilized various machine learning models with the ground-based MEP air quality dataset. For example, Li et al. (2017b) utilized a long short-term memory (LSTM) neural network extended model to predict $PM_{2.5}$

concentrations at a maximum forecast horizon of 24 h for air quality monitoring stations in Beijing, China. Wu et al. (2020) proposed a composite prediction system based on an LSTM neural network to predict daily $PM_{2.5}$/$PM_{10}$ in Wuhan City. Zhang et al. (2020) established a hybrid model integrating deep learning with multi-task learning to predict hourly $PM_{2.5}$ concentrations in three different districts of Lanzhou City. Ke et al. (2021) utilized four machine learning models to develop an air quality forecasting system that can automatically find the best "model and hyperparameter" combination for the next 3-day

air quality forecast in seven megacities of China. These works are highly valuable in exploring novel methods of air quality prediction relative to conventional CTMs. To the best of our knowledge, the aforementioned studies on air pollution forecasting solely focused on a few monitoring stations, typical cities, or small regions, while national-level air quality predictions remain lacking. The challenges in national-level forecasting include substantial temporal and spatial variances (Song et al., 2017b) in air pollution and enormous computational power requirements.

The curse of dimensionality is a common obstruction in modeling, i.e., an increasing amount of input data leads to rapidly increasing complexity, and prediction algorithms are susceptible to over-fitting (Rodriguez-Galiano et al., 2012). Therefore, considerable research has focused on reducing the dimensionality of input data by selecting only significant variables and eliminating redundancy. The methods of this research can be classified into three categories: the filter method (e.g., a correlation matrix using the Pearson Correlation), wrapper method (e.g., recursive feature elimination), and embedded method (e.g.,

Lasso regularization) (Chandrashekar and Sahin, 2014). These methods can reduce the adverse effects of irregular variables or noise while retaining prediction performance (Guyon and Elisseeff, 2003). They also save computing resources for model training. However, in previous studies on air quality forecasting, filter methods such as Pearson correlation coefficients or the maximal information coefficient (MIC) (Kinney and Atwal, 2014) were commonly utilized for input selection. These input selection methods can help improve the performance of machine learning models; however, they all have serious limitations.

For example, universal meteorological variables that highly correlate with $PM_{2.5}$ in a large region of China are difficult to find using Pearson correlation coefficients because they can vary substantially both spatially and temporally (Zhai et al., 2019). MIC is the most employed method for capturing linear and nonlinear correlations between variable pairs (Chen et al., 2016). However, it cannot consider relevance and redundancy simultaneously (Sun et al., 2018). Furthermore, MIC is computationally intensive (Cao et al., 2021).

Machine learning algorithms are often considered "black-box" models that learn the input–output relationship from immense training samples (Casalicchio et al., 2019). Many researchers have devoted enormous efforts to developing and implementing tools to interpret machine learning models. Among these tools, game-theoretic formulations of feature significance are the most widely utilized because they can capture the interactions among features (Shapley, 1952), and they may be the only solution satisfying the four "favorable and fair" axioms (Fryer et al., 2021). Several scholars have conducted in-depth studies

on distinguishing feature significance based on the Shapley value (Shapley, 1952). For example, Park and Park (2021) utilized





the Shapley additive explanation (SHAP) approach (Lundberg and Lee, 2017a) to interpret multiple machine learning models and found that most of the models have similar features. Golizadeh Akhlaghi et al. (2021) successfully interpreted the feature contributions of the Guideless Irregular Dew Point Cooler on the predicted parameters based on SHAP. In addition to SHAP, which explains individual predictions, Covert et al. (2020) proposed a novel method that can explain model behavior across

the entire dataset (global interpretability), called Shapley additive global importance (SAGE). SHAP and SAGE both utilize the Shapley value; however, compared with SHAP, SAGE can simultaneously eliminate larger subsets of redundant features (Covert et al., 2020). Additionally, SAGE extracts features from the conditional distribution instead of the marginal distribution because the latter may lead to breaking feature dependencies and producing unlikely feature combinations (Lundberg and Lee, 2017b). Furthermore, investigating the feature importance based on model performance (Jothi et al., 2021) has been verified

as a meaningful and effective approach for interpreting data-driven models and is popular in computer science (Altmann et al., 2010). However, this method has rarely been applied to air quality forecasting using machine learning tools.

       In the present study, the first version of a regional feature selection-based machine learning system (RFSML v1.0) is developed. The system can predict short-term air quality with high accuracy in China. In this study, the RFSML system predicts the primary air pollutant ($PM_{2.5}$) concentration over every target site from the China MEP air quality monitoring network by

learning its implicit trend from long-term series records. This method can be extended to other airborne pollutant predictions in future studies. SAGE analysis is adopted to interpret valuable features and exclude redundant inputs to avoid over-fitting the model during training. Because the SAGE calculations are more time consuming than the model training, as explained in Section 3.1, they are not repeated for every target site but are implemented in limited ensemble sites that are randomly selected in a given region. China was divided into five densely populated regions, according to the air pollution patterns, which are

consistent with the Clean Air Action target regions released by the Chinese State Council, as discussed in Section 2.2.3. The top three critical features in the ensemble SAGE calculations were utilized as the input features for the implicit trend model training for each site. The robustness of the regional feature selection was tested over three widely utilized machine learning models, i.e., the random forest (RF), gradient boosting (GB), and multi-layer perceptron (MLP) models, and four forecasting horizons (6, 12, 18, and 24 h).

25       The remainder of the paper is organized as follows: The composition of the data used in this study and the pre-processing method are introduced in Section 2. Then, the three machine learning models and their hyperparameter choices utilized in this study are described. The principles of SAGE and the details of the SAGE ranking-based regional feature selection are described at the end of Section 2. In Section 3, the computational costs of SAGE and machine learning model training are detailed. Then, the results of feature selection in each region are presented and analyzed. The prediction performance of RFSML is evaluated

and compared with that of the standard machine learning process. Finally, the conclusions and future prospects are provided in Section 4.





## 2 Model, data, and methods

The components of the RFSML method that are used to forecast $PM_{2.5}$ concentrations are described in the following sections.

### 2.1 Model domain and data

5    The RSFML system forecasts air pollution levels in the vicinity of a monitoring station. This forecasting uses machine learning by examining the variability in the available station datasets. The monitoring network consisted of 1588 stations, for data collected in 2019, at the locations displayed in Fig. 1. Because the station network is dense, pollution level forecasting can be performed for nearly any location in eastern China.

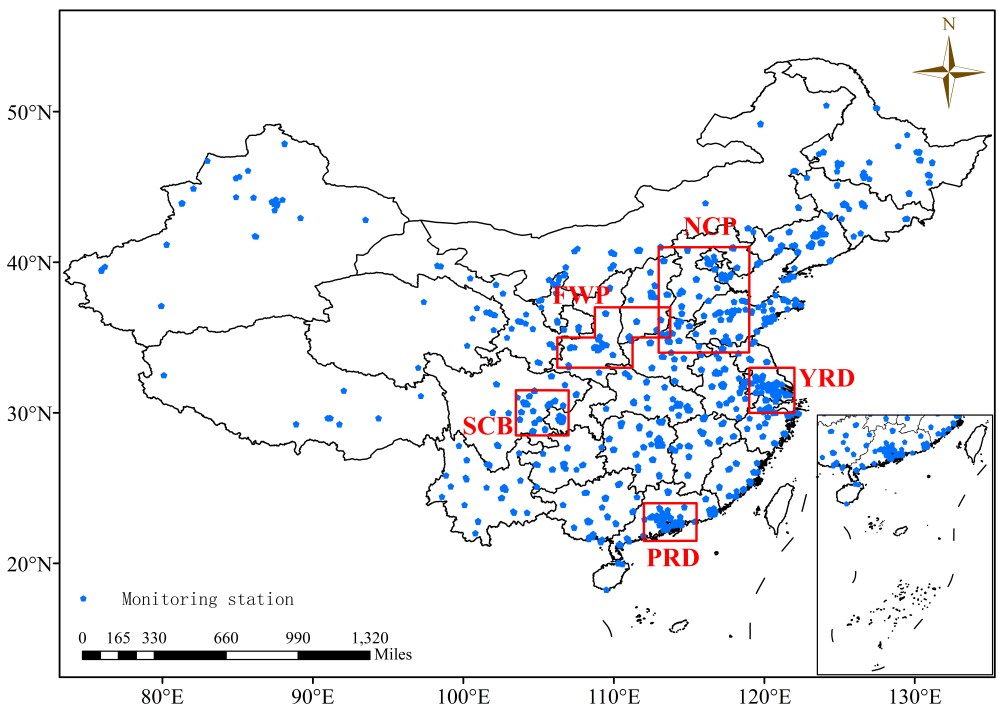

**Figure 1.** Locations of environmental monitoring stations in China in 2019 (blue pentacles). Red rectangles represent the five primary mega-city clusters in China.

The input data for machine learning consisted of hourly averaged air pollutant measurements (e.g., $PM_{2.5}$, CO, $SO_2$, and
10    $NO_2$) from the Chinese MEP monitoring network, meteorological reanalysis data from ERA5-Land (Muñoz Sabater et al., 2021), and atmospheric composition data from the Copernicus Atmosphere Monitoring Service (CAMS) reanalysis (Inness et al., 2019) provided by the European Centre for Medium-Range Weather Forecasts (ECMWF), and emission data from the Multi-resolution Emission Inventory for China (MEIC) inventory with time factors applied at an hourly resolution. The input





data are summarized in Table 1. The variables in the datasets are correlated with and may drive the PM$_{2.5}$ concentration and are therefore useful predictors.

Data from 2018 and 2019 were used in the experiments. The first 15,690 h (from January 1, 2018 to October 15, 2019) were used for model training and cross-validation, and the actual tests were performed using the remaining 1824 hours of data

from October 15, 2019 to December 30, 2019.

**Table 1.** Summary of empirical input variables

| Type | Amount | Spatial esolution | Temporal resolution | Source |
|---|---|---|---|---|
| Ground observation | 4 | monitoring station | Hourly | Monitoring station |
| atmosperic composition | 4 | 0.75° x 0.75° | 3-Hourly | CAMS global reanalysis |
| Meteorology | 8 | 0.1° x 0.1° | Hourly | ERA5-Land hourly data |
| MEIC | 9 | 0.25° x 0.25° | Monthly | MEIC v1.3 |
| Time factor | 2 | | Hourly | |

### 2.1.1   Air pollutant observations

The observed air pollutant concentrations at the stations were used as inputs (NO$_2$, SO$_2$, and CO) and target variables (PM$_{2.5}$) in the model training. The available time series of PM$_{10}$ observations was missing many values and was therefore excluded from the model. Additionally, O$_3$ observations were excluded because these data exhibit a diurnal cycle that substan-

tially differs from the PM$_{2.5}$ target concentrations.

Missing data occurred for each of the studied pollutants because of equipment failure, incorrect sensor readings, and improper operation. For the PM$_{2.5}$ time series, approximately 14.6% of the observations were missing on average, as illustrated in Fig. 2(a). However, an uninterrupted time series is necessary for model training and rolling forecasting. In studies such as (Qin et al., 2019; Ma et al., 2019a), it was shown that the observations from surrounding monitoring stations can be utilized

to insert suitable values for missing data through imputation. Data imputation tools, such as cubic interpolation, have gained popularity for enhancing monotone data (Fritsch and Carlson, 1980).

In studies such as (Qin et al., 2019; Ma et al., 2019a), it was shown that the observations from surrounding monitoring stations can be utilized to insert suitable values for missing data through imputation. Data imputation tools, such as cubic interpolation, have gained popularity for monotone data (Fritsch and Carlson, 1980).

In this study, the K-Nearest Neighbor (KNN) classification method (Zhang, 2012) and cubic imputation were combined to create an uninterrupted time series. The KNN algorithm is illustrated as Algorithm 1 in the Supplement. The KNN algorithm was implemented using the following steps:

1. Monitoring stations with over 20% missing data were excluded from the training and prediction because such large amounts of missing data are not believed to be filled with sufficient accuracy.





2. For each station, the number of monitoring stations within a radius of 0.8°was calculated (following the empirical choices suggested in (Jin et al., 2019)). If fewer than three surrounding stations were available, the station was excluded from the training and forecasting. If three or four surrounding stations were found, these were all selected, while four stations were selected randomly if more than four surrounding stations were found.

5  3. For each target station, a geographic inverse distance weighting technique (Bartier and Keller, 1996) was used to estimate the missing values using the observed values from the surrounding stations.

After KNN interpolation, the amount of missing data in the PM$_{2.5}$ time series was reduced to approximately 4.5%, as illustrated in Fig. 2(b).

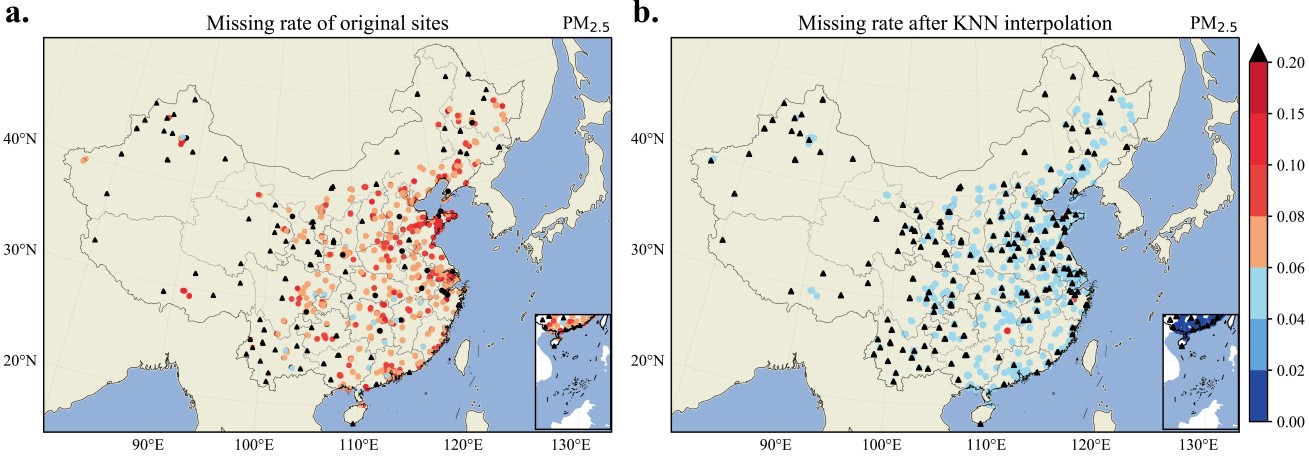

**Figure 2.** Missing fraction of (a) original and (b) KNN-interpolated PM$_{2.5}$ data. Dots and triangles denote the locations of air quality monitoring stations, and dot colors represent the missing data rate of each monitoring station. Black triangles indicate monitoring stations with 20% missing data or over 15% missing data after KNN interpolation that were excluded from the model.

Because there were cases where nearby stations and the target station simultaneously exhibited missing data, some in-
10  stances of missing values remained after KNN interpolation. Therefore, cubic imputation (Kincaid et al., 2009) was employed to insert values for the remaining missing data. Outliers generated by the cubic imputation were replaced with the minimum or maximum of the original series. A total of 1263 monitoring stations exhibited no missing data after applying interpolation. Both the mean and standard deviation of the homogenized time series were similar to those of the original data, as illustrated in Fig. S1 of the Supplemental Material.

### 2.1.2 Air pollutant forecast product & meteorological variables

The CAMS reanalysis (Inness et al., 2019) provides three-dimensional simulations of the atmospheric composition obtained by combining a global atmospheric chemistry model and observations. Therefore, it is expected to surpass pure model-





based prediction accuracy. Selected concentrations of trace gasses and aerosols from the CAMS reanalysis were inputs for the PM$_{2.5}$ predictor. The PM$_{2.5}$ simulations in this dataset were also used as a benchmark for the RFSML prediction.

We obtained the 3-hourly reanalysis data of four air pollutant concentrations (pm2p5, no2, so2 and co), which are reanalysis data (pm2p5, no2, so2 and co) of four ground observations mentioned above, in China from 2018 to 2019. Cubic imputation

was applied to fill in the missing data because the 3 h resolution of the CAMS reanalysis data is too coarse for interpolation. After interpolation, $0.75^o$ x $0.75^o$ grid data were imputed to each monitoring station.

Meteorological variables, as illustrated in Table 2, were obtained from ERA5-Land data ( Copernicus Climate Change Service (C3S) , 2017) at a horizontal resolution of $0.1^o \times 0.1^o$ degrees and an hourly temporal resolution for 2018 and 2019. The data are available from the Climate Data Store via DOI:10.24381/cds.e2161bac. The data were interpolated to the monitoring

station locations for use in machine learning.

**Table 2.** Summary of meteorological variables obtained from ERA5-land dataset

| Meteorology | Long name | Unit |
|---|---|---|
| u10 | 10m u-component of wind | m·s$^{-1}$ |
| v10 | 10m v-component of wind | m·s$^{-1}$ |
| d2m | 2m dew-point temperature | K |
| t2m | 2m temperature | K |
| skt | Skin temperature | K |
| sp | Surface pressure | Pa |
| tp | Total precipitation | m |
| str | Surface net thermal radiation | J·m$^{-2}$ |

### 2.1.3 Emission inventory

MEIC, the most popular anthropogenic emission inventory in China (Li et al., 2017a), has been validated to provide consistent aerosol precursor loading for satellite observations (Fan et al., 2018). It has been widely employed to quantify the air pollution in multi-atmosphere chemical models. The latest inventory of 2017 from MEIC version 1.3 was obtained

via *http://meicmodel.org/index.html* (last access: October 2021) for use in this study. Based on the emission source height distribution, 24 h distribution, and Regional Acid Deposition Model, version 2 (Zimmermann and Poppe, 1996) chemical reaction scheme, the original monthly emission data were processed into hourly emission rates. Considering their correlation with PM$_{2.5}$, nine pollutant species were selected as machine learning predictor inputs, as displayed in Table 3.





**Table 3.** Summary of emission inventory variables

| MEIC | Full name |
|---|---|
| E_CO | CO |
| E_ECI | Elemental Carbon $PM_{2.5}$-nuclei mode |
| E_HCHO | Formaldehyde |
| E_NH3 | $NH_3$ |
| E_NO2 | $NO_2$ |
| E_ORGJ | Organic $PM_{2.5}$-accumulation mode |
| E_PM25J | Unspeciated primary $PM_{2.5}$-accumulation mode |
| E_PM_10 | Unspeciated Primary $PM_{10}$ |
| E_SO2 | $SO_2$ |

## 2.2 The RFSML system

### 2.2.1 System framework

Fig. 3 displays the framework of the proposed RFSML and a standard machine learning system. Note that a standard machine learning system refers to a machine learning system without any feature selection. Standard machine learning is conducted as follows. First, all observations and datasets related to $PM_{2.5}$ evolution are collected, and then the missing values are interpolated into the original dataset. Next, the appropriate machine learning model is selected, and the continuous data time series is reformed into the required input structure. The model is then trained repeatedly until the appropriate hyperparameters are obtained, and finally, predictions are made with the trained model. Given an input $x_n$ that consists of individual features

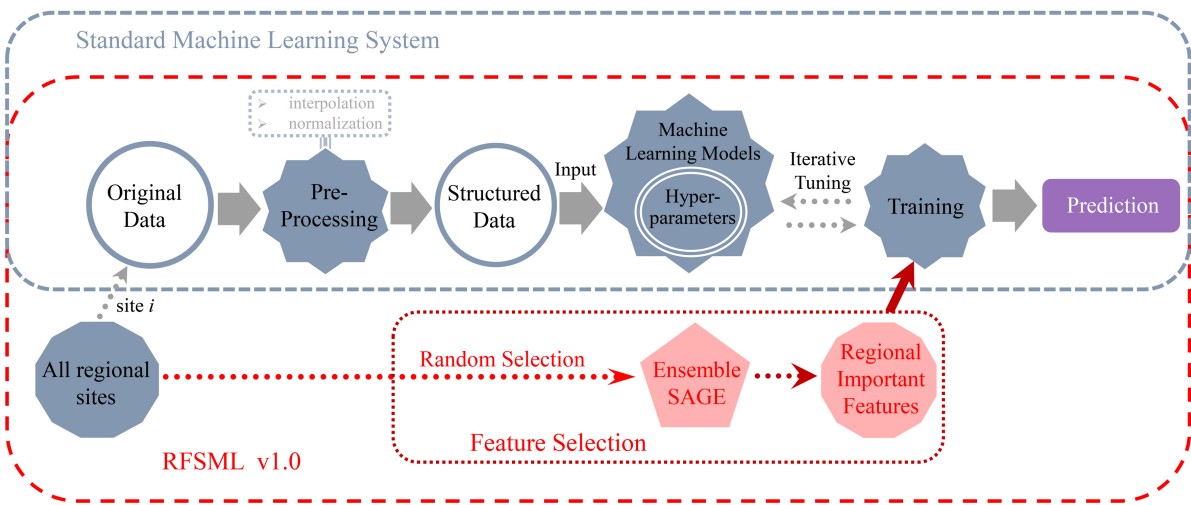

**Figure 3.** RFSML versus standard machine learning system framework.





$(x_1, x_2, ..., x_n)$, a predictor $\mathcal{F}$ is utilized in a supervised learning task to predict the target variable $y$. A time series regression, such a rolling forecast, can be expressed as follows:

$$\hat{y}_{t+h} = \mathcal{F}(x_1^{t-t_p+1}, \cdots, x_1^t, x_2^{t-t_p+1}, \cdots, x_2^t, \cdots, \cdots, x_n^{t-t_p+1}, \cdots, x_n^t) \tag{1}$$

where, at any instant $t$, the input vector storing $n$ individual features in the previous $t_p$ h is utilized to forecast the target PM$_{2.5}$

concentration $\hat{y}$ with a horizon of $h$ h. The forecast predictor $\mathcal{F}$ represents the machine learning model (RF, GB, or MLP) trained using the historical data. Details on the selection of $t_p$ and $h$ are provided in Section 2.2.2.

As aforementioned, some features are residual, and the feature subset can provide sufficient predictive power and less noise for $\mathcal{F}$. Thus, the proposed RFSML utilized SAGE to obtain the optimal feature subsets. Considering the computational efficiency, we divided the total national air quality monitoring stations into six types, each of which randomly ensembled the air

quality monitoring stations for feature selection. Given any feature subset $\boldsymbol{x}_s = \{x_1, x_2, ..., x_s\}$, the machine learning models can be described as follows:

$$\hat{y}_{t+h} = \mathcal{F}(x_1^{t-t_p+1}, \cdots, x_1^t, x_2^{t-t_p+1}, \cdots, x_2^t, \cdots, \cdots, x_s^{t-t_p+1}, \cdots, x_s^t) \tag{2}$$

### 2.2.2   Machine learning models

Different machine learning algorithms have been used to forecast PM$_{2.5}$ because they can provide promising approaches to

handle complex nonlinear relationships. Each algorithm exhibits advantages and drawbacks. Of the machine learning models, typical boosting (e.g., GB) and bagging (e.g., RF) algorithms are widely applied in regression analysis by using a set of decision trees. Additionally, artificial neural network models (e.g., MLP) that are composed of many processing elements can successfully perform nonlinear mapping. Thus, to evaluate the robustness of the feature selection, all of the prediction algorithms mentioned above were tested in the present study.

The original data in Fig. 3 were converted into a 27-dimensional matrix ($n = 27$) after preprocessing. On the basis of the auto-correlation and partial auto-correlation results, time step $t_p = 9$ h was selected for the forecast. Predicting horizon $h$ spans from 1 h to 24 h. Then, the matrix was converted into supervised learning based on $t_p$ and $h$. The model hyperparameters (Table 4) were designed for each predicting algorithm using 10-fold cross-validation and then fit to each predicting algorithm.

**Table 4.** Summary of model's hyperparameters

| GB | | RF | | MLP | |
|---|---|---|---|---|---|
| Hyperparameter | Final Value | Hyperparameter | Final Value | Hyperparameter | Final Value |
| N estimators | 100 | N estimators | 100 | Neurons in hidden layer | 100 |
| Max depth | 3 | Max depth | None | Activation | Relu |
| Loss | Mean square error | Loss | Mean squared error | Loss | Mean squared error |
| Learning rate | 0.1 | Min samples leaf | 1 | Solver | Adam |



### 2.2.3 SAGE-based regional feature selection

Air pollution in nearby monitoring stations has inherent similarities because their forcing factors, i.e., meteorological and emission variables, are closely related in a given region. As in Zhai et al. (2019), all the available sites were partitioned into six categories in the present study: the North China Plain (NCP; 34–41°N, 113–119°E), Yangtze River Delta (YRD; 30–33°N,

119–122°E), Pearl River Delta (PRD; 21.5–24°N, 112–115.5°E), Sichuan Basin (SCB; 28.5–31.5°N, 103.5–107°E) Fenwei Plain (FWP; 33–35°N, 106.25–111.25°E; 35–37°N, 108.75–113.75°E), and the remainder of China. The locations of these regions can be found in Fig. 1.

Many methods have been utilized to investigate the significance of features for machine learning models. The game-theoretic method based on the Shapley value is the most widely adopted. Unlike SHAP, a well-known method for explaining

individual predictions, SAGE explains model behavior across the entire dataset. Global model interpretability helps us understand the distribution of target outcomes based on the features (Molnar, 2019), which is useful for finding the typical features of each region. There are two outstanding added values of SAGE (Covert et al., 2020). The first is its ability to remove large subsets of features because only removing individual features gives too little significance to features with sufficient proxies, such as in permutation tests. The other advantage of SAGE is its ability to select notable features from their conditional distribution

instead of their marginal distribution, reducing unlikely feature combinations.

Given the function $W_{\mathcal{F}}$, which represents the predictive power of a machine learning model $\mathcal{F}$ with subsets of features $\boldsymbol{x}_s \subseteq \boldsymbol{x}_n$, the SAGE algorithm can be written as follows:

$$W_{\mathcal{F}}(S) = -\mathbb{E}[\, \ell(\, \mathbb{E}[\, \hat{y} \mid \boldsymbol{x}_s \,], y \,) \,] \tag{3}$$

$$\phi_i(W_{\mathcal{F}}) = \frac{1}{n} \sum_{S \subseteq N \setminus \{i\}} \binom{n-1}{|S|}^{-1} \big[ W_{\mathcal{F}}(S \cup \{i\}) - W_{\mathcal{F}}(S) \big] \tag{4}$$

where $\ell$ means the loss function that measures the root mean squared error (RMSE) or mean absolute error (MAE); $\hat{y}$ is the prediction from $\mathcal{F}$; $y$ represents the target variable; sets $S$ and $N$ store $\{1, 2, 3, \cdots, s\}$ and $\{1, 2, 3, \cdots, n\}$, respectively; $i$ is each single variable where $i \in N$; $n$ is the length of N. $W_{\mathcal{F}}$ increases with a decline in the loss function for any subset $S \subseteq N$ (note the minus sign in front of the loss function in Equation 3). Equation 4 represents the Shapley value that is the weighted

average of the incremental changes from adding $i$ to subsets $S \subseteq N \setminus \{i\}$ (Covert et al., 2020). The more a feature contributes to the prediction from $\mathcal{F}$, the larger the positive values $\phi_i(W_{\mathcal{F}})$ would become.

The framework of regional feature selection, as illustrated by algorithm 1, is as follows. Fifteen ensemble stations are randomly selected in each of the six regions. Taking NCP as an example, the significance of the features of the ensemble monitoring stations with four prediction horizons (6, 12, 18, and 24 h) and three prediction algorithms are analyzed using

SAGE algorithms. Then, the outcomes of the ensemble-SAGE model are ranked, as displayed in the heatmap in Fig. 4. The heatmap highlights the significant features. PM$_{2.5}$, CO, and v10 typically exhibit higher ranks in the 15 random monitoring stations and four prediction horizons. The heatmaps of the ensemble-SAGE analyses of the other five regions can be found in



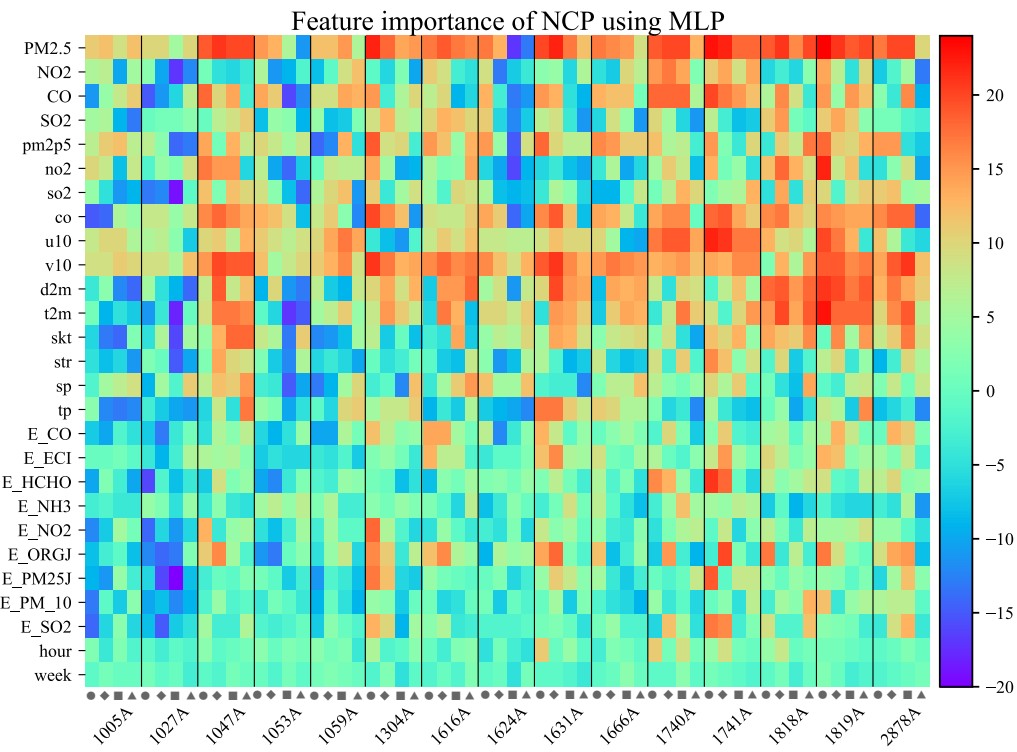

**Figure 4.** Heatmap of all empirical features with 15 random monitoring stations in NCP and four prediction horizons. The circle, diamond, square, and triangle represent the 6, 12, 18, and 24 h prediction horizons, respectively. The heatmap is based on the SAGE analysis ranking of features training by MLP. The warmer the row color, the more significant the corresponding feature.

Supplemental Material Figs. S2-18. The feature significances in different regions are ranked by the sum of the SAGE values in the ensemble monitoring stations and four prediction horizons, as displayed in Fig. 5. The feature selection based on the ensemble-SAGE analysis concerning Fig. 5 is explained in Section 3.2.

# 3 Results and discussion

## 3.1 Computation complexity analysis

Instead of performing feature selection for every forecast model independently, our proposed ensemble-SAGE analysis successfully interprets the important regional features for PM$_{2.5}$ prediction with substantially less computation complexity. In addition, the regional feature selection improves the forecast accuracy and saves significant computing power for the machine learning model training by excluding redundant inputs and speeding up the model convergence. In this study, all computations





---

**Algorithm 1** Regional feature selection

---

**Input:** data $\{\text{site}_d\}_{d=1}^{D}$, region $z$, machine learning model $\mathcal{F}$, predicting horizon $h$, SAGE algorithm

1: Initialize $h = [6, 12, 18, 24]$, ensemble size = 15

2: **for** $j = 1$ **to** $len(z)$ **do**

3:     Find all sites ($D_r$) in $z_j$ from $\{\text{site}_d\}_{d=1}^{D}$

4:     Select ensemble sites from $D_r$ randomly

5:     **for** $e = 1$ **to** ensemble size **do**

6:       **for** $f = 1$ **to** $len(\mathcal{F})$ **do**

7:         **for** $g = 1$ **to** $len(h)$ **do**

8:           Employ SAGE algorithm

9:           Rank importance of input ($A$) for each $h$, $\mathcal{F}$ and $z$

10:         **end for**

11:         Re-rank $A$'s importance ($B$) for each $\mathcal{F}$ and $z$

12:       **end for**

13:     **end for**

14:     Re-rank $B$'s importance ($C$) for each $z$

15:     Take the three most important variables as features for each z

16: **end for**

---

**Table 5.** Summary of mean and maximum time costs of machine learning model training

| Time (s) | Mean_standardML | Mean_RFSML | Max_standardML | Max_RFSML |
|:---:|:---:|:---:|:---:|:---:|
| **GB** | 91.024 | 12.57 | 93.952 | 13.751 |
| **RF** | 291.466 | 36.854 | 319.75 | 42.907 |
| **MLP** | 5.592 | 3.553 | 18.316 | 12.237 |

concerning the SAGE-based feature selection and machine learning model training were conducted on several nodes configured with 4 x 16-core 2.1 GHz Intel Xeon E5-2620 v4 CPUs and with 64 GB of memory.

The computational cost of SAGE varies significantly, with average times of 7353, 3891, and 3325 s when using GB, RF, and MLP, respectively. The maximum time costs reach 61,209; 65,127; and 23,931 s with GB, RF, and MLP, respectively.
5   Thus, using SAGE for each PM$_{2.5}$ forecasting model with different air quality monitoring stations, prediction horizons, and machine learning models is time consuming. As illustrated in Table 5, the time cost for the three machine learning model training sessions is greatly reduced using the inputs from SAGE-based regional feature selection.

## 3.2  Regional feature selection analysis

The results of the SAGE-based regional feature selection concerning the three machine learning models, six partitioned
10   regions, and four forecasting horizons are discussed in this section. Taking the NCP as an example, critical features that govern



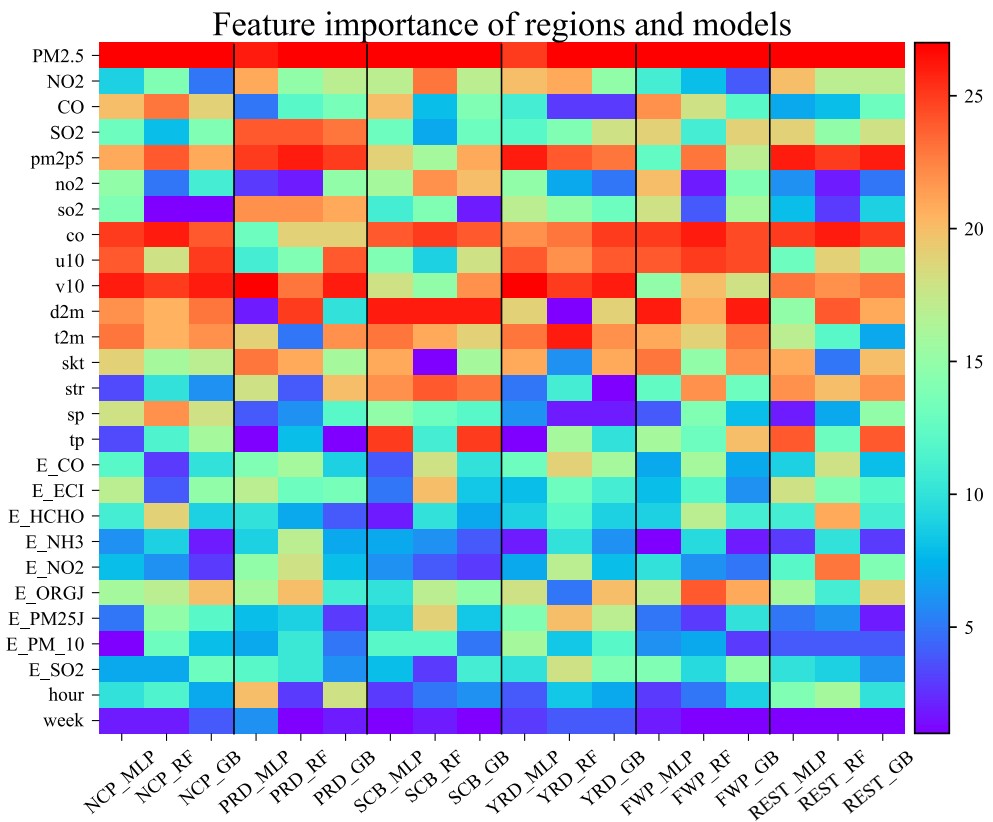

**Figure 5.** Heatmap of empirical features for six regions with three machine learning models. Each column represents the rearrangement of the sum of 15 monitoring stations and 4 prediction horizons. Black vertical lines are used to distinguish each region. The warmer the row color, the more critical the corresponding feature.

the performance of PM$_{2.5}$ forecasting vary across stations and prediction horizons, as illustrated in Fig. 4. However, PM$_{2.5}$, CO, and v10 (features) play an overwhelmingly positive role in MLP-based PM$_{2.5}$ forecasting for most selected stations and predicting horizons. This result indicates that these three features suit all of the stations in the NCP. SAGE analysis heatmaps for other regions or using different prediction algorithms can be found in Figs. S2-S18 of the Supplemental Material. Consistently

5    critical features can be easily extracted from the five mega-city cluster regions using our selection method. However, they are difficult to extract from the remaining area of China. This area does not have universal key features because its stations are spread widely across China and therefore exhibit substantially different air quality patterns. An improved station clustering method can help solve this issue and will be explored in our future research.

      To extract the important robust features that fit all stations in a given region, we summed and ranked the SAGE analysis

10    values in the ensemble monitoring stations and prediction horizons. The ensemble-SAGE ranking is displayed in Fig. 5. There are consistent, crucial features in the six cluster regions regardless of the prediction algorithm or horizon. PM$_{2.5}$ is the most



**Table 6.** Summary of selected features

| Region | NCP | PRD | SCB | YRD | FWP | REST |
|---|---|---|---|---|---|---|
| | PM$_{2.5}$ | PM$_{2.5}$ | PM$_{2.5}$ | PM$_{2.5}$ | PM$_{2.5}$ | PM$_{2.5}$ |
| **Feature** | v10 | v10 | d2m | v10 | d2m | co |
| | co | pm2p5 | tp | pm2p5 | co | pm2p5 |

critical feature for predicting its trend at a particular region and with a particular prediction algorithm. In addition to PM$_{2.5}$, two variables from CAMS reanalysis, co and pm2p5, are critical across all regions. This result suggests that the forecast of these variables from CAMS reanalysis can help capture the varying trend in the machine learning models, even though the predictions are different from the actual values. By contrast, time factors (week and hour information) are the least important

features for short-term prediction. This result is consistent with that of Hu et al. (2014), where no distinct weekday/weekend difference was observed for PM$_{2.5}$ in the NCP and PRD.

Considering their generality and robustness, we selected the top three critical features for each region, as illustrated in Table 6. Note that the ensemble SAGE analysis selected different key features in different regions. In the NCP, the simulation of CO from CAMS reanalysis, which is a representative air pollutant, includes valuable information other than CO observations.

This result implies that local precursor emissions are a major contributor to PM pollution (Guo et al., 2016), and non-point source pollution may be more favorable for PM$_{2.5}$ forecasting. Additionally, v10, which represents regional transmission, is a critical feature for PM$_{2.5}$ forecasting in the NCP, PRD, and YRD. This result indicates that regional transmission plays a vital role in those three regions (Chen et al., 2017; Liu et al., 2017). This finding is consistent with those reported in recent studies. Zhang et al. (2018) found that the anomalously high, normalized, and near-surface meridional wind is typically the primary

cause of the severe haze in the NCP using a chemical transport model. Huang et al. (2018) illustrated that regional transport accounts for over half of PM$_{2.5}$ under the polluted northerly airflow in winter. Ma et al. (2019b) discovered that the regional PM$_{2.5}$ pollution in winter is primarily from North and East China using a trajectory model. However, v10 is less significant in the SCB. This result is because of the blocking effect of the plateau terrain on the northeasterly winds (Shu et al., 2021); hence, winds are frequently static, particularly in winter and autumn (Liao et al., 2017). By contrast, d2m and tp are crucial features

for hourly PM$_{2.5}$ forecasting in the SCB. This finding may be because polluted weather patterns are typically associated with higher relative humidity in that area, and tp, representing rainfall, is vital to eliminate air pollution in a basin (Zhan et al., 2019).

### 3.3   Performance of RFSML

This section presents the forecasting skill of the proposed RFSML system driven by regional features selected by the

ensemble-SAGE-based model. The results are also compared with those of a standard machine learning forecasting model and fourth-generation ECMWF global reanalysis data. The latter is referred to as the benchmark of chemical transport models.





Fig. 6 displays the times series of the simulated PM$_{2.5}$ for the three forecasting systems (MLP model and a predicting horizon of 12 h) versus observational data. Each subplot represents a random monitoring station in the corresponding five megacity cluster regions. The subplots illustrate the typical behaviors observed for the other monitoring stations, machine learning models, and prediction horizons. Both the standard machine learning and RFSML models outperform the simple

**Figure 6.** Time series of test time in five mega-city cluster regions. The black dots and red pentacles represent original and interpolated PM$_{2.5}$ respectively. The solid lines with color of dim gray, light sky blue and dark violet represent prediction of CAMS reanalysis, standard machine learning system and RFSML respectively. Panel a, b, c, d and e represent a random site in NCP, YRD, PRD, SCB and FWP respectively. Note that those are 12 hours in advance prediction which are in parallel with CAMS reanalysis's predicting horizon and the machine learning model used here is MLP.





CTM model. This result indicates that the machine learning algorithms are superior in air pollution prediction (Pérez et al., 2000). Additionally, PM$_{2.5}$ predictions with selected key features perform better than the standard machine learning forecast that uses all related features. The GB and RF machine learning models used in this study also show steady improvements.

Both the RFSML and standard machine learning predictions typically underestimate high PM$_{2.5}$ concentrations as the prediction horizon increases. This underestimation can be ascribed to three primary possible reasons. First, the correct features are difficult to obtain, and unsuitable features can bring significant bias and noise into the prediction algorithms. Second, the construction of our prediction algorithm network may be insufficiently complex or deep to determine the actual relationship between features and the PM$_{2.5}$. However, considering the purpose of a real-time forecast, the time to forecast, which is closely related to the complexity of the prediction algorithm network, cannot be too long. Third, considering our test period

only included late autumn and early winter of 2019, the training and validation periods only included autumn and winter of 2018, which are too short for a prediction algorithm to learn the complex relationship for hourly PM$_{2.5}$ forecasting. Seasonal training and validation may obtain satisfactory outcomes for a particular seasonal forecast (Bai et al., 2019).

Fig. 7 displays the spatial distribution of the RMSEs (columns a and b) and MAEs (columns c and d) of the PM$_{2.5}$ forecast for all stations either using the standard machine learning or RFSML system at a forecasting horizon of 12 h. The RMSEs and

MAEs significantly decreased when using the selected key features for all three machine learning models, particularly in regions with severe PM$_{2.5}$ pollution, e.g., the NCP and FWP. This consistent improvement also occurs when the forecast horizon changes to 6, 18, and 24 h, and the results are illustrated in Figs. S19-21 of the Supplemental Material.

A modified Taylor diagram (Taylor, 2005) is plotted in Fig. 8 to show the overall outcome. RFSML forecasts with selected features typically exhibit a lower RMSE and higher R than the standard forecasts. The best improvement is obtained when the

deep learning (MLP) model is used, while forecasts with the selected new features in the RF model are not significantly improved and even not as good as with forecasts that use all features.

This result can be explained by the characteristics of the two types of prediction algorithms. RF increases the diversity of the trees through the bootstrapped aggregation of several regression trees (bagging) (Brokamp et al., 2017). It has the advantage of maintaining low bias because tree-based methods with bagging can reduce the variance of an estimated prediction function.

Some uninformative features can be ignored through bagging, i.e., RF reduces the high variance by growing the individual trees to a deep level and then making their predictions, typically through averaging (Liaw et al., 2002). By contrast, MLP, which implements the global approximation strategy (Osowski et al., 2004), may face problems of multicollinearity and noise caused by uninformative features.

The RMSE increases and R declines with an increase in the prediction horizons across all regions and machine learning

models in general. The average coefficient of determination (R$^2$) of the 24 h forecast (the maximum horizon set in this study) based on the three machine learning models increases from 0.47 to 0.65 in the NCP, from 0.41 to 0.52 in the PRD, from 0.62 to 0.67 in the SCB, from 0.44 to 0.57 in the YRD, and from 0.62 to 0.65 in the FWP when using the ensemble-SAGE analysis-based feature selection. This results indicate that the RFSML system can provide the operational PM$_{2.5}$ forecast with a maximum horizon of 24 h.



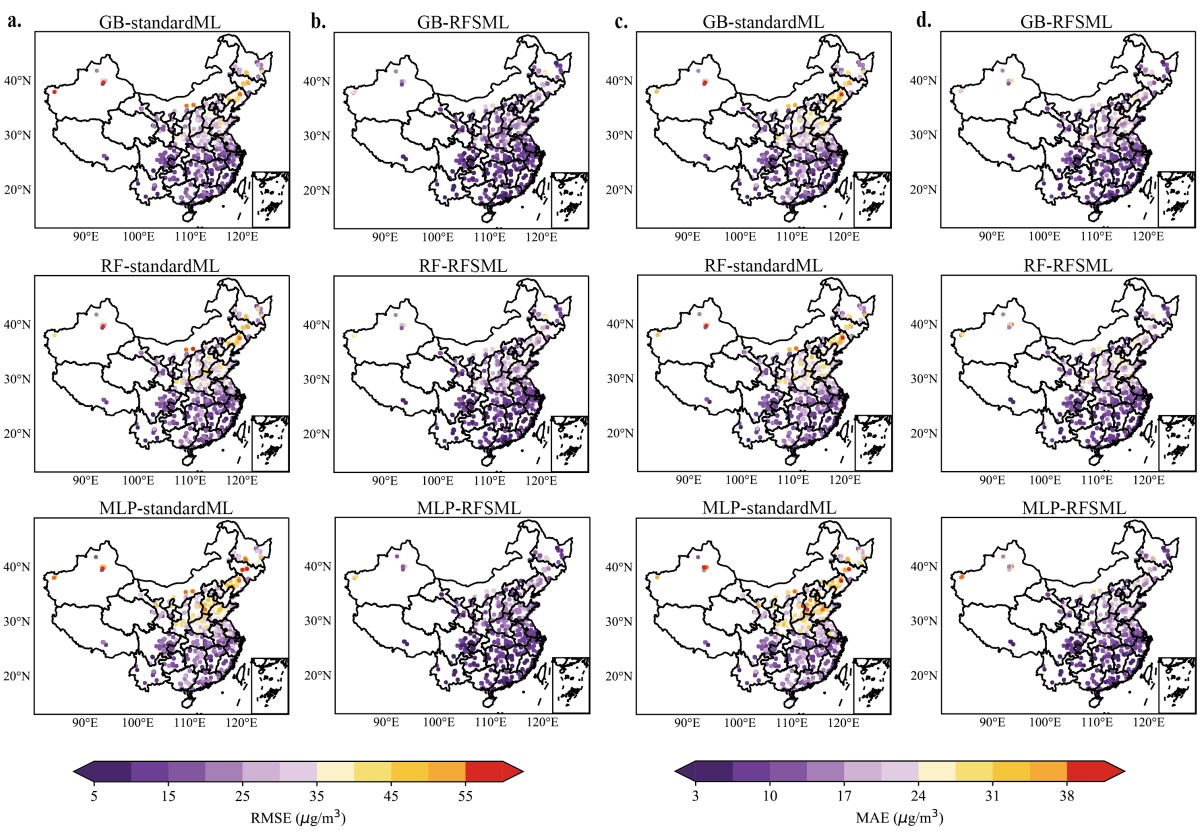

**Figure 7.** Spatial distribution of RMSEs and MAEs at a prediction horizon of 12 h. Panels a and c are results of standard machine learning system, while panels b and d are results of RFSML. The cooler the color tone, the lower the RMSEs and MAEs, and thus the better the prediction performance.

## 4   Conclusions and future work

Machine learning models have been successfully utilized in air quality forecasts worldwide because of their high computational efficiency and accuracy. However, substantial room for improvement remains. In this study, we developed the RFSML v1.0 system, which can predict national air quality with high accuracy in real time in China.

5   In a standard machine learning system, all related features are typically utilized in model training and prediction. However, the high dimensionality and redundant input data may lead to increased complexity and machine learning model over-fitting. To overcome this obstacle, we combined an ensemble-SAGE analysis with our RFSML system. This method extracts the key features in a given region at an affordable extra cost, and the significance of those regional selected features are explained physically. Compared with the standard machine learning system that was fed with all relative features, the RFSML system driven

10   by the selected key features resulted in superior interpretability, less training time, and more accurate predictions. Statistically, the average RMSE and MAE of predictions were reduced from 24.74 and 16.54 $\mu g/m^3$ to 21.54 and 13.7 $\mu g/m^3$, respectively,



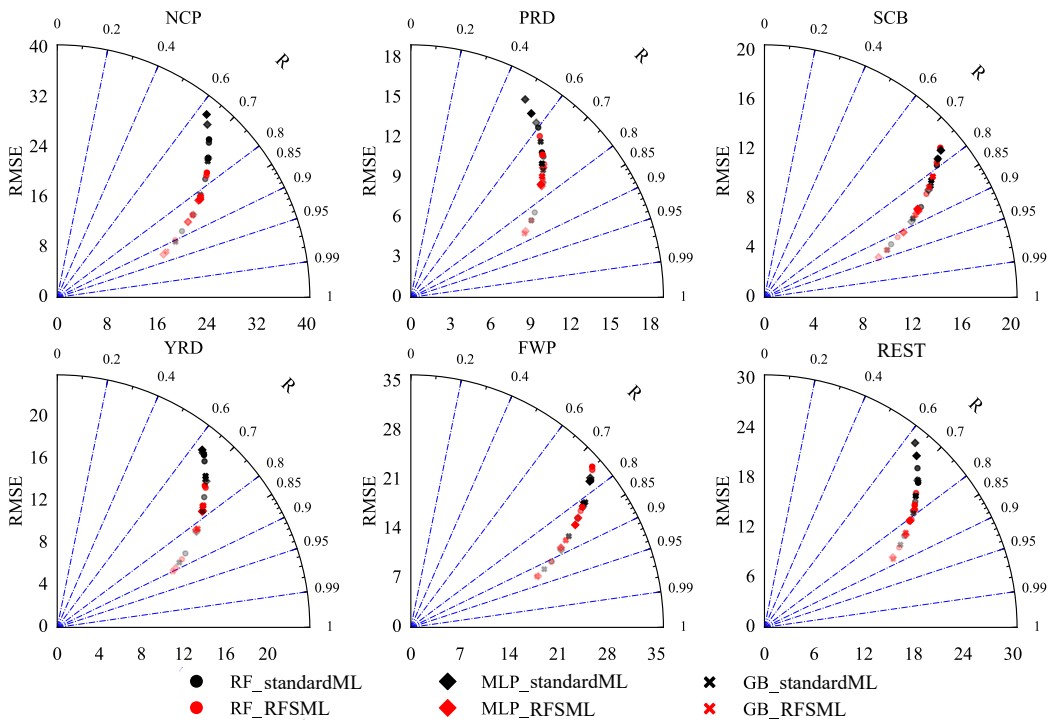

**Figure 8.** A modified Taylor Diagram that illustrates RMSE and correlation coefficient in six regions. The black and red colors represent forecast with standard machine learning models and RFSML. Round, diamond and fork represent RF, MLP and GB respectively. The transparency of markers indicates the four prediction horizons, where the transparency increases as the forecast hours increase.

with RFSML. Additionally, $R^2$ increased from 0.6 to 0.7, and the average forecasting model training cost was reduced from 129.36 s to 17.66 s. Among the three machine learning models studied, the prediction performance of RFSML with MLP exhibited the greatest increase, with $R^2$ increasing from 0.55 to 0.72. By contrast, RF exhibited the least improvement, with $R^2$ increasing from 0.61 to 0.66. In addition, RF and GB were more robust than MLP for certain underlying uninformative

5    features, while MLP was more susceptible to over-fitting.

The six-region partition used here was not based in science. Additionally, stations in a given region may exhibit different air quality patterns, particularly in the "REST" region. Therefore, our ensemble-SAGE analysis does not always select the representative feature, limiting the machine model interpretability and prediction ability. A more scientific station partition should be determined for future studies.

10    Based on the results of this study, the RFSML system can accurately predict air quality in the short term at the national scale; this renders it valuable for health professionals and policy makers in terms of providing early warning to population





categories more susceptible to air pollution (e.g., children, elderly, and people with respiratory or cardiovascular issues) and in reducing and regulating air pollution.

**Code and data availability**

The ground-based air quality monitoring observations are from the network established by the China Ministry of Environ-
mental Protection and accessible via https://quotsoft.net/air/, the measurements used in this study also are archived on Zenodo (https://doi.org/10.5281/zenodo.6551820; Li Fang, 2022). The RFSML algorithm is in the Python environment and is archived on Zenodo (https://doi.org/10.5281/zenodo.6551850; Li Fang, 2022).

**Acknowledgments**

This work is supported by the National Natural Science Foundation of China [grant 42105109] and Natural Science
Foundation of Jiangsu Province (NO.BK20210664).

**Author contribution**

JJ and LF conceived the study and designed RFSML system. LF wrote the code of RFSML and carried out the predic-
tion and evaluation. HXL, AS, CX, TD and HL provided useful comments on the paper. LF prepared the manuscript with contributions from JJ and all others co-authors.

**Competing interests**

The authors declare that they have no conflict of interest.



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
