# Peer review of "Development of a regional feature selection-based machine learning system (RFSML v1.0) for air pollution forecasting over China"

_Geoscientific Model Development, 2022_

## Author Comment (AC1)

**Response to Referee #1:** We would like to thank the referee for the careful review throughout the paper and the in-depth comments that help to improve our paper.

Our Reply follows (*the reviewer's comments are in italics and blue*)

*General Comments*

*The paper is quite interesting, but I still have some comments on it.*

*Major comments*

*Q1: how could you identify the improvement comes from your new methods or just spliting data to 6 groups? spliting data into spatial groups also can help model more easier to capture the variation.*

**Reply:** Thanks for the comments and this was indeed not clearly explained in our previous version. Actually the model training and predicting are performed on each site independently. We could also simply perform the SAGE for calculating the key features in any given sites independently, however, this is very expensive as has been described in ***Section. 3.1 Computational complexity analysis***. Meanwhile, Air pollution in nearby monitoring stations has inherent similarities because their forcing factors, i.e., meteorological and emission variables, are closely related in a given region. Therefore, all sites are divided into six clusters, according to the air pollution patterns, which are consistent with the Clean Air Action target regions released by the Chinese State Council. For a given group, SAGE analysis is implemented in 15 randomly selected sites for having the key features. These features would then be used in the training and prediction in every regional site.

To explain this, remarks are now added in page 4, line 16-20 "***SAGE analysis is adopted to interpret valuable features and exclude redundant inputs to avoid over-fitting the model during training. Because the SAGE calculations are more time consuming than the model training, as explained in Section 3.1, they are not repeated for every target site but are implemented in limited ensemble sites that are randomly selected in a given region***", and in page 11, line 21-30 "***The computational costs of the SAGE analysis over machine learning models including RF, GB and MLP are presented in Section 3.1. They are much more expensive than the model training therefore cannot be repeated over all sites. Meanwhile, air pollution in nearby monitoring stations has inherent similarities because their forcing factors, i.e., meteorological and emission variables, are closely related in a given region. As in Zhai et al. (2019), all the available sites were partitioned into six categories in the present study: the North China Plain (NCP; 34–41◦N, 113–119◦E), Yangtze River Delta (YRD; 30–***

*33◦N, 119–122◦E), Pearl River Delta (PRD; 21.5–24◦N, 112–115.5◦E), Sichuan Basin (SCB; 28.5–31.5◦N, 103.5–107◦E) Fenwei Plain (FWP; 33–35◦N, 106.25–111.25◦E; 35–37◦N, 108.75–113.75◦E), and the remainder of China. The locations of these regions can be found in Fig. 1. Therefore, we propose the regional future selection in which SAGE are only implemented in limited ensemble sites that are randomly selected in a given region, and the selected features would be used for model training and predicting in each regional site.*", and in page 16, line 12-14 "*To highlight the improvements by using the selected key features, the regional performance which represents the average of the forecasting performance in all sites of the given region is introduced.*".

*Q2: why you split the sites into 6 categories? and when you apllied your final models, how could you define the predicted location/grid belong to which categories?*

**Reply:** It is believed that air pollution in nearby monitoring stations has inherent similarities because their forcing factors, i.e., meteorological and emission variables, are closely related in a given region. Therefore, we spilt all sites into six clusters, according to the air pollution patterns, which are consistent with the Clean Air Action target regions released by the Chinese State Council. The categories mainly help us to obtain the regional important features with less computation power.

Of course, this method of division is empirical and a little coarse. Spatial clustering methods that are based on time series analysis would be considered in our future study. Remarks are added in page 20, line 21-24 by saying "*The six-region partition used here was empirical and not based on science. Additionally, stations in a given region may exhibit different air quality patterns, particularly in the "REST" region. Therefore, our ensemble-SAGE analysis does not always select the representative feature, limiting the machine model interpretability and prediction ability. A more scientific station partition like spatial clustering would be determined for future studies.*"

*Q3: please add the spatial cross-validation results to check your model spatail predict ability*

**Reply:** Thanks for the comments. The model training is carried out in each of the observing sites independently, in which the historical records are available. Therefore, we only provide prediction over these sites instead of a grided one, and cross validation is not necessary in this paper.

Note that we are exactly exploring for a full prediction that covers the whole model domain from this current work RFSML. The basic ideas of is to fuse the high-quality RFSML prediction and the gridded CTM prediction with larger uncertainty using Bayesian Theory. The diagram can be found in the Figure below. Here the blue lines represent the high-quality forecast available at several single stations, and the model is trained using the observations marked by black dots; the blue face here denotes the chemical transport model (CTM) giving the gridded forecast which is however usually biased. The RFSML and CTM prediction can be considered as two estimates of future situation, and each of them has the weakness and advantage. Bayesian theory will be used to fuse them together, and resulting a gridded and less-biased forecast like the brown face. That work will be soon submitted as a companion paper with this RFSML work.

[Figure]

Figure Diagram of a gridded prediction from the RFSML prediction at single stations.

Remarks are now added in the ***Conclusion and future*** by saying "***Meanwhile, RFSML provides only predictions over the air quality monitoring sites where historical data is available for machine learning model training, instead of a grided forecast. A Bayesian theory -based prediction fusion is being explored now to extend the RFSML forecast available at single stations to a gridded one.***" in page 20, line 18-20.

*Q4: for the temporal validation, .the test data are only in winter? have you tried to use rolling temopral validation? use previous 4 seasons as training, 1 season as validation. The study only use 2 year data, I wonder whether model can be predicted in the following year. the study period did not include 2020, so will the model forecast be affected by covid-19 when we applied this model for the real early warning system? Have you test the models with 2020 or 2021? this will be the important issue for early warning system.*

**Reply:** Thanks for point this issue. It is because winter is the most severe-polluted season therefore we chose it as the test period in this paper. It is also necessary to point out that our

waring system can be operated in a rolling way. To validate the forecast sill, we make an extra prediction over 2020 April (during COVID-19) with the model trained by the recent two year's data. Promising results are obtained as well.

Remarks are now added in page 6, line 5-6 for explaining the rolling forecast test "***Our RFSML system can of course operate in a rolling way, additional forecasts in a less-polluted and emergency period 2020 April are performed with the models are trained using the recent two-year data similarly.***", and in page 19-20, line 9-11 and line 1-2 by saying "***To further confirm the predicting capability in a rolling way, we make forecasts over a less polluted month April 2020. Specific results can be found in Supplemental Material Tables S1. Steady improvement of predicting performance is still achieved by RFSML. Time series as given in Figure S22 show similar result as main text that RFSML has better predict ability than standard machine learning. As is illustrated in Figure S23-24, RFSML has both lower RMSE and MAE than standard machine learning, which implies the advantage of RFSML.***"

*Table S1. Summary of prediction performance in the time period of April, 2020.*

| Region | Metric | Predicting horizon | | | |
| | | 6 | | 18 | |
| | | standardML | RFSML | standardML | RFSML |
|---|---|---|---|---|---|
| NCP | RMSE | 17.71 | 12.2 | 22.11 | 16.71 |
| | MAE | 14.06 | 9.3 | 17.86 | 13.19 |
| | R | 0.71 | 0.83 | 0.5 | 0.69 |
| PRD | RMSE | 10.7 | 7.78 | 13.17 | 11.1 |
| | MAE | 8.51 | 5.74 | 10.38 | 8.39 |
| | R | 0.83 | 0.9 | 0.7 | 0.77 |
| SCB | RMSE | 13.29 | 10.37 | 17.02 | 13.51 |
| | MAE | 10.13 | 7.63 | 13.11 | 10.2 |
| | R | 0.72 | 0.81 | 0.53 | 0.66 |
| YRD | RMSE | 14.08 | 10.43 | 18.67 | 14.41 |
| | MAE | 11.27 | 8.09 | 14.76 | 11.48 |
| | R | 0.75 | 0.87 | 0.51 | 0.74 |
| FWP | RMSE | 16.26 | 13.24 | 19.8 | 16.44 |
| | MAE | 12.69 | 10.14 | 15.65 | 12.97 |
| | R | 0.66 | 0.73 | 0.47 | 0.6 |
| REST | RMSE | 21.59 | 17.89 | 26.01 | 22.25 |
| | MAE | 14.29 | 10.5 | 17.48 | 13.62 |
| | R | 0.68 | 0.79 | 0.48 | 0.66 |

[Figure]

***Figure S22. Time series of a prediction horizon of 6 hours in five mega-city cluster regions. The black dots and red pentacles represent original and interpolated PM2.5 respectively. The solid lines with light sky blue and dark violet represent prediction of standard machine learning system and RFSML respectively. Panel a, b, c, d and e represent a random site in NCP, YRD, PRD, SCB and FWP respectively.***

[Figure]

*Figure S23. Spatial distribution of RMSE in a predicting horizon of 6 and 18 hours. Panel a and c are results of standard machine learning system while panel b and d are results of RFSML. The cooler the color tone, the lower the RMSE, thus the better predicting performance.*

[Figure]

*Figure S24. Spatial distribution of MAE in a predicting horizon of 6 and 18 hours. Panel a and c are results of standard machine learning system while panel b and d are results of RFSML. The cooler the color tone, the lower the MAE, thus the better predicting performance.*

*Q5: the ground-level pollutant is also included as input? how could you predict the value for thoes location outside these sites?*

**Reply:** Answer could be found in the ***Reply*** to *Q3*.

***Minor comments***

**Reply:** Here "***None***" in Scikit learn means "nodes are expanded until all leaves are pure or until all leaves contain less than min_samples_split samples". We added an annotate for it in ***Section 2.2.2 Machine learning models*** on page 10 line 21-22 by saying "***Note that None for the max depth of RF means "nodes are expanded until all leaves are pure or until all leaves contain less than min_samples_split samples" in Scikit learn (Pedregosa et al., 2011).***"

*why each region choose 15 sites for features selection? have you tried 10,20, or some other numbers?*

**Reply:** We randomly choose 15 sites with four predicting hours and three machine learning models, which is a total of 180 samples. However, the time cost for SAGE for each sample is quite large. We also tried 20 sites for feature selection and the top three most important features are almost the same. Choice of 10 sites results rather inconsistent selection. So, we choose 15 as it is not that time consuming but representative enough. Explanation was added in page 12, line 7-9 "***Note that we also tried randomly ensemble numbers 10 and 20 in NCP key feature extraction using the MLP model at several prediction horizons. The choice of 15 shows to give the robust result with the minimum computation cost, and it is therefore used for all regional feature selections in this study.***"

*why do you pick only top 3 features? not 5 or more? will they affect the results?*

**Reply:** We believe the number for top important features will affect the result to some extent. In our case, the top 3 features are significantly more important than other candidate features, and some robustness are lost with more input features as our region division is not good enough. Generally, we suggest less features for a large region to guarantee the robustness while more features for a small region.

***Technique points:***

*data were imputed to each monitoring station" here you mean impute the missing or downscale? if downscale, from 0.75 to which resolution?*

**Reply:** Sorry for the confusion. We first impute the 3 h resolution of CAMS reanalysis data into 1 h resolution by cubic imputation. Then the hourly data is imputed to each site by finding the nearest grid. We will reexplain it in ***Section 2.1.2 Air pollutant forecast product & meteorological variables*** by replacing "" with "***The 3 h temporal resolution of the CAMS reanalysis data is firstly interpolated into 1 h resolution by cubic imputation. Then continuous time series of features at the monitoring stations are extracted from the interpolated 1h data at a resolution of $0.75^o$ x $0.75^o$ using the nearest mapping.***" in page 8, line 2-5.

*"The data were interpolated to the monitoring station locations for use in machine learning." It is still confusing.*

**Reply:** We now explained it in ***Section 2.1.2 Air pollutant forecast product & meteorological variables*** by replacing "." with "***The time series of meteorological variables used for the machine learning are extracted from this product using the nearest mapping method.***" in page 8, line 8-9.

**References**

Pedregosa, F., Varoquaux, G., Gramfort, A., Michel, V., Thirion, B., Grisel, O., Blondel, M., Prettenhofer, P., Weiss, R., Dubourg, V., Vanderplas, J., Passos, A., Cournapeau, D., Brucher, M., Perrot, M., and Duchesnay, E.: Scikit-learn: Machine Learning in Python, Journal of Machine Learning Research, 12, 2825–2830, 2011.

---

## Author Comment (AC2)

**Response to Referee #2:** We would like to thank the referee for the careful review throughout the paper and the useful comments.

Our Reply follows (*the reviewer's comments are in italics and blue*)

*General Comments*

*This paper presents the development of an improved machine learning based air quality nowcasting system. Instead of using all possible related features in the model training and predicting, they selected those general important and effective features. The feature selection is done using a computationally efficient ensemble method. Their nowcasting system is tested on the PM2.5 forecast on a national scale and validated to be superior than a CTM model and conventional MLs. Generally speaking, the paper is clearly written and well structure, their results are scientifically solid. I recommend accepting it after a minor revision. I also have questions and comments for the author that could help to improve their manuscript.*

*Major comments*

*They have tested their regional feature selection-based ML nowcast system at a national scale and using several common ML models (RF, GB and MLP), which makes their results very sound. However, it is only tested at a 2019 winter season. I understand winter is the most severe polluted season there. Their system should be able to operate in a rolling forecast way. If extra training and testing are conducted at a less-polluted period/season, this study would be an excellent paper.*

**Reply:** We thank the reviewer for the very comment and agree with the point. We made extra forecast in April 2020 which is a less-polluted month. The new experimental results reproduce the advantages of RFSML and we believe this experiment enriches the results of the original paper.

Remarks concerning the rolling forecast tests are now added in page 6, line 5-6 "***Our RFSML system can of course operate in a rolling way, additional forecasts in a less-polluted and emergency period 2020 April are performed with the models are trained using the recent two-year data similarly.***", and in page 19-20, line 9-11 and line 1-2 by saying "***To further confirm the predicting capability in a rolling way, we make forecasts over a less polluted month April 2020. Specific results can be found in Supplemental Material Tables S1. Steady improvement of predicting performance is still achieved by RFSML. Time series as given in Figure S22 show similar result as main text that RFSML has better predict ability than standard machine learning. As is illustrated in Figure S23-24, RFSML has both lower RMSE and MAE than***

*standard machine learning, which implies the advantage of RFSML.*"

**Table S1. Summary of prediction performance in the time period of April, 2020.**

| Region | Metric | Predicting horizon | | | |
|--------|--------|------------|-------|------------|-------|
| | | 6 | | 18 | |
| | | standardML | RFSML | standardML | RFSML |
| NCP | RMSE | 17.71 | 12.2 | 22.11 | 16.71 |
| | MAE | 14.06 | 9.3 | 17.86 | 13.19 |
| | R | 0.71 | 0.83 | 0.5 | 0.69 |
| PRD | RMSE | 10.7 | 7.78 | 13.17 | 11.1 |
| | MAE | 8.51 | 5.74 | 10.38 | 8.39 |
| | R | 0.83 | 0.9 | 0.7 | 0.77 |
| SCB | RMSE | 13.29 | 10.37 | 17.02 | 13.51 |
| | MAE | 10.13 | 7.63 | 13.11 | 10.2 |
| | R | 0.72 | 0.81 | 0.53 | 0.66 |
| YRD | RMSE | 14.08 | 10.43 | 18.67 | 14.41 |
| | MAE | 11.27 | 8.09 | 14.76 | 11.48 |
| | R | 0.75 | 0.87 | 0.51 | 0.74 |
| FWP | RMSE | 16.26 | 13.24 | 19.8 | 16.44 |
| | MAE | 12.69 | 10.14 | 15.65 | 12.97 |
| | R | 0.66 | 0.73 | 0.47 | 0.6 |
| REST | RMSE | 21.59 | 17.89 | 26.01 | 22.25 |
| | MAE | 14.29 | 10.5 | 17.48 | 13.62 |
| | R | 0.68 | 0.79 | 0.48 | 0.66 |

[Figure]

***Figure S22. Time series of a prediction horizon of 6 hours in five mega-city cluster regions. The black dots and red pentacles represent original and interpolated PM2.5 respectively. The solid lines with light sky blue and dark violet represent prediction of standard machine learning system and RFSML respectively. Panel a, b, c, d and e represent a random site in NCP, YRD, PRD, SCB and FWP respectively.***

[Figure]

*Figure S23. Spatial distribution of RMSE in a predicting horizon of 6 and 18 hours. Panel a and c are results of standard machine learning system while panel b and d are results of RFSML. The cooler the color tone, the lower the RMSE, thus the better predicting performance.*

[Figure]

*Figure S24. Spatial distribution of MAE in a predicting horizon of 6 and 18 hours. Panel a and c are results of standard machine learning system while panel b and d are results of RFSML. The cooler the color tone, the lower the MAE, thus the better predicting performance.*

*They should also explain the current machine learning model cannot fully replace model-based air quality forecasting systems, as ML models could not be trained and operated without inputs from the historical measurements. While for many rural regions, they are unavailable. The authors should explain this point clear.*

**Reply:** We agree with referee that the ML cannot fully replace the current "causation" model that is a parameterization of physical rules in nature, while ML is purely based on data correlations. On the other hand, the current RFSML can indeed provide forecast at single stations instead of a full gridded one.

Actually we are exploring for a full prediction that covers the whole model domain from this current work RFSML. The basic ideas of is to fuse the high-quality RFSML prediction and the gridded CTM prediction with larger uncertainty using Bayesian Theory. The diagram can be found in the Figure below. Here the blue lines represent the high-quality forecast available at several single stations, and the model is trained using the observations marked by black dots;

the blue face here denotes the chemical transport model (CTM) giving the gridded forecast which is however usually biased. The RFSML and CTM prediction can be considered as two estimates of future situation, and each of them has the weakness and advantage. Bayesian theory will be used to fuse them together, and resulting a gridded and less-biased forecast like the brown face. That work will be soon submitted as a companion paper with this RFSML work.

[Figure]

Figure Diagram of a gridded prediction from the RFSML prediction at single stations.

Remarks are now added in the Conclusion and future by saying "***Meanwhile, RFSML provides only predictions over the air quality monitoring sites where historical data is available for machine learning model training, instead of a grided forecast. A Bayesian theory -based prediction fusion is being explored now to extend the RFSML forecast available at single stations to a gridded one.***" in page 20, line 18-20.

*Minor comments*

*Page 6, Table 1: esolution to resolution*

**Reply:** Corrected.

*Page 12, line 5: computational complexity?*

**Reply:** Corrected.

*Page 17, line 1: and at a predicting horizon?*

**Reply:** Corrected.